# Optimal Piecewise Polynomial Approximation for Minimum Computing Cost by Using Constrained Least Squares

**DOI:** 10.3390/s24123991

**Published:** 2024-06-20

**Authors:** Jieun Song, Bumjoo Lee

**Affiliations:** Department of Electronic Engineering, Myongji University, Yongin 17058, Republic of Korea; song12069595@gmail.com

**Keywords:** piecewise polynomial, function approximation, regression, constrained least squares

## Abstract

In this paper, the optimal approximation algorithm is proposed to simplify non-linear functions and/or discrete data as piecewise polynomials by using the constrained least squares. In time-sensitive applications or in embedded systems with limited resources, the runtime of the approximate function is as crucial as its accuracy. The proposed algorithm searches for the optimal piecewise polynomial (OPP) with the minimum computational cost while ensuring that the error is below a specified threshold. This was accomplished by using smooth piecewise polynomials with optimal order and numbers of intervals. The computational cost only depended on polynomial complexity, i.e., the order and the number of intervals at runtime function call. In previous studies, the user had to decide one or all of the orders and the number of intervals. In contrast, the OPP approximation algorithm determines both of them. For the optimal approximation, computational costs for all the possible combinations of piecewise polynomials were calculated and tabulated in ascending order for the specific target CPU off-line. Each combination was optimized through constrained least squares and the random selection method for the given sample points. Afterward, whether the approximation error was below the predetermined value was examined. When the error was permissible, the combination was selected as the optimal approximation, or the next combination was examined. To verify the performance, several representative functions were examined and analyzed.

## 1. Introduction

### 1.1. Importance of Approximation and Previous Studies

In many applications in the scientific and engineering fields, approximation for a complex function or data set is important, such as compression of ECG signals [1]; various voice processing applications such as speech recognition, synthesis, and conversion [2,3]; and correction of sensor data [4,5]. Numerous methods have been studied for approximation, with the least squares method being the preferred choice. The least squares method offers high accuracy by minimizing the residuals and is also simple to implement in a computer program. Formulating the optimal coefficients as least squares problems dates back to Gendre (1805) and Gauss (1809). The first application was performed by Gergonne in 1815 [6]. Therefore, in this paper, the least squares method is employed for approximation.

The approximation error at the sample points generally tends to decrease as high-order polynomials are used, but this causes overfitting, which leads to significant oscillation between sample points. To resolve the overfitting, it is recommended to use lower-order polynomials by splitting the sections. In the case of using piecewise polynomials, it takes less computation time than using a single high-order polynomial with the same approximation error. Since the maximum number of intervals and orders depends on the number of samples and the complexity, several studies have been proposed to find the appropriate order and number of intervals. In [7,8,9,10], piecewise polynomial fitting was proposed. Additionally, ref. [7] proposed least squares piece fitting using a cubic polynomial, ref. [9] used a method of adjusting the boundary of the segments to increase the operation speed, and ref. [10] used piecewise polynomial modeling to lower the error rate. However, in [8,9,10], there was a limitation that the order of the polynomial must be determined by the user. In [4], the whole domain was evenly divided into several intervals, and each interval was approximated by a cubic polynomial using the least squares method with the constraint that it had continuity in all boundaries. The studies cited in [11,12,13,14,15] proposed an approximation method using piecewise linear approximation (PLA), while ref. [16] proposed a method consisting of using several linear functions and then moving the endpoints of the interval appropriately to reduce the approximation error of the interval.

There are two considerations when approximating with piece-wise polynomials: the order of polynomials and the number of intervals. Higher-order polynomials require higher computational costs. Further, this may cause overfitting. Therefore, it is necessary to determine the appropriate order of the polynomials. When using approximated function, i.e., piece-wise function, it requires additional steps to determine the subdomain corresponding to the input value. If there are many intervals, the approximation precision increases, but more time is also needed to determine the subdomain. Thus, the order and the number of polynomials should be balanced and optimized. In order to accomplish this, an optimization scheme is proposed in contrast with [4,5,6,7,8,9,10,11,12,13,14,15,16], which utilized a predetermined polynomial order and number of polynomials.

Several studies have adopted optimization methods for curve fitting. Among the optimization methods, the hp-adaptive pseudo-spectral method, proposed in [17], is most similar to the algorithm proposed in this paper. The method determines the locations of segment breaks and the order of the polynomial in each segment. To be more specific, when the error within a segment displays a consistent pattern, the number of collocation points is increased. On the other hand, if there are isolated points with significantly higher errors than the rest of the segments, then the segment is split at those points. This produces solutions with better accuracy than global pseudo-spectral collocation while utilizing less computational time and resources. The hp-adaptive pseudo-spectral method and the optimal piecewise polynomial (OPP) approximation algorithm proposed in this paper are similar in that they increase accuracy by increasing the number of orders and intervals. The OPP approximation algorithm compares computational costs to obtain an approximation of the minimum cost while satisfying the error norm constraint for the approximation.

### 1.2. OPP Approximation Algorithm

As mentioned in the introduction, polynomial approximation methods are necessary for sensor approximation and other approximation tasks. Through a literature review concerning such methods, we propose the optimization of piecewise polynomials. As an improvement over previous approximation methods, the proposed OPP approximation algorithm automatically determines both the order and number of intervals while focusing on computational efficiency to obtain the approximate polynomial with the smallest computational cost within the specified error norm.

As seen in Figure 1, approximation is processed by finding the optimized order and number of intervals that satisfy the error norm constraint with the minimum computational cost. To achieve this, the algorithm adopts two cost functions. One is for the approximation error that is used by the least squares. The calculated approximation error is compared with the error norm constraint for approximation. The other is computational cost function. It indicates the program run time needed to calculate the value of the function for an input value, x, and is composed of two runtime costs: the cost of polynomial function call and the cost of a binary search tree to determine the relevant interval. In other words, the computational cost is calculated given the order of the polynomial and the number of intervals. The costs for all possible combinations are calculated and sorted in ascending order offline. This is used to search for the order and number of intervals satisfying the error constraint with the minimum cost. Therefore, the OPP approximation algorithm is useful in systems that must be efficient to compute, such as real-time systems and embedded systems. When approximating, if the approximation functions of each interval are placed independently, discontinuity occurs over the entire interval. To address this, a constraint is introduced using the Lagrange multiplier method to smoothly connect the approximation functions of each interval. Applying the approximation function obtained using the OPP approximation algorithm to the systems can minimize the computational cost while satisfying the error norm constraint. In this way, obtaining an approximation function before applying the system can reduce computational time and memory.

This paper is organized as follows. In Section 2, previous knowledge related to the algorithms proposed Section 3 is described. In Section 4, the algorithm is examined with representative test functions. Subsequently, the OPP approximation algorithm is discussed, and further work to improve the performance is proposed in Section 5.

## 2. Preliminaries

### 2.1. Nomenclature

The algorithm and all formulas in this study use the notation in Table 1. The symbols necessary to understand the algorithm are as follows: $a_{k,i}$ means the coefficient of the $i^{th}$ order term of the $k^{th}$ polynomial. $q_{k}$ represents the coefficients of the approximation polynomial in the $k^{th}$ interval as a vector, and q is a (m+1)n vector concatenated as $q_{k}$. $c_{a}$ is the approximation cost and $c_{c}$ is the computational cost at runtime. $m^{*}$ and $n^{*}$ indicate the optimal polynomial order and the number of intervals, respectively. * describes the optimized value for $c_{a}$ within $\epsilon_{a}$ and the minimum $c_{c}$. Scalars, vectors, and matrices are written in lowercase, lowercase boldface, and uppercase boldface, respectively.

### 2.2. Formulation of Constrained Least Squares

The least squares method is a representative approach in regression analysis to approximate functions or discrete samples by minimizing the sum of the squares of the residuals in the results of each individual equation. The residual is the difference between the given sample and the approximate value, represented by weighted squares, as shown in (1). The matrix form of the least squares method is represented as follows:(1)$$e=\frac{1}{2}\sum_{k=1}^{n}\left({\left(y_{k}-F_{k}q_{k}\right)}^{T}W_{k}(y_{k}-F_{k}q_{k})\right)$$
where $F_{k}=\left[\begin{array}{ccccc} 1 & x_{k,1} & x_{k,1}^{2} & \cdots & x_{k,1}^{m}\\ 1 & x_{k,2} & x_{k,2}^{2} & \cdots & x_{k,2}^{m}\\ \vdots & \vdots & \vdots & \ddots & \vdots \\ 1 & x_{k,\eta \left(k\right)} & x_{k,\eta \left(k\right)}^{2} & \cdots & x_{k,\eta \left(k\right)}^{m} \end{array}\right]$, $y_{k}={\left[\begin{array}{ccc} y_{k,1} & \cdots & y_{k,\eta \left(k\right)} \end{array}\right]}^{T}$, and $q_{k}={\left[a_{k,0}a_{k,1}\cdots a_{k,m}\right]}^{T}$. The least squares method is accomplished by finding q, which makes the partial differential of error, i.e., the gradient, 0. Consequently, this leads to optimal polynomials with the minimum average of the residual sum of squares at the sample points. For the sake of simplicity, Equation (1) can be shortened into a single monomial expression using the sparse diagonal matrix F as follows:(2)$$e=\frac{1}{2}{\left(y-Fq\right)}^{T}W\left(y-Fq\right),$$
where $F=diag\left(F_{k}\right),y={\left[\begin{array}{ccc} y_{1}^{T} & \cdots & y_{n}^{T} \end{array}\right]}^{T}$, and $q={\left[\begin{array}{cc} q_{1}^{T} & \begin{array}{cc} \cdots & q_{n}^{T} \end{array} \end{array}\right]}^{T}$. The polynomial, $p_{k}$, should be smooth at the boundaries, $v_{k-1}$ and $v_{k}$, with adjacent polynomials $p_{k-1}$ and $p_{k+1}$, respectively. Without a loss of generality, in the proposed algorithm, the 1st-order differentiability was appended as a constraint, Gq=0, for the smoothness. Note that if necessary, it is possible to increase the order for differentiability at the intersection of the adjacent intervals. Consequently, Equation (2) was modified with the Lagrange Multiplier, λ, as follows:(3)$$e=\frac{1}{2}{\left(y-Fq\right)}^{T}W\left(y-Fq\right)+\lambda^{T}G,$$
where $G=\mathrm{diag}\left(G_{k}\right)$ and $G_{l}=\left[\begin{array}{cccccccc} 1 & v_{k} & \cdots & v_{k}^{m} & -1 & -v_{k} & \cdots & -v_{k}^{m}\\ 0 & 1 & \cdots & mv_{k}^{m-1} & 0 & -1 & \cdots & -mv_{k}^{m-1} \end{array}\right]$. By partial differentiation of (3) with respect to q and λ, two equations are obtained, i.e., $\frac{\partial }{\partial q}e=0$ and Gq=0. By solving the above equation, the optimal coefficients, which minimize $c_{a}$, are obtained.
(4)$$q=H(-G^{T}{\left(GHG^{T}\right)}^{-1}GHF^{T}W^{T}y+F^{T}W^{T}y),$$
where $H={\left(F^{T}W^{T}F\right)}^{-1}$.

## 3. OPP Approximation Algorithm

### 3.1. Overall Algorithm Flow

The proposed algorithm is intended for use in systems where computing time is critical, such as real-time systems and/or embedded systems. With the algorithm’s runtime, it is particularly important to reduce function call times. Since the approximate function is composed of several polynomials, it must be known how much time is required to execute the function at runtime. To accomplish this, the computational cost function was defined from the four arithmetic operations and binary search tree. This is explained in more detail in the next subsection. Offline, based on the computational cost function, an approximation polynomial with the smallest calculation time was obtained, then used in the system.

The overall flow for the OPP approximation is described in Figure 2. First, the maximum values of polynomial order, $m_{max}$, and the number of intervals, $n_{max}$, to explore were set. Afterward, the computational costs for all possible combinations of the number of intervals and the order were calculated and tabulated in ascending order ($m\in \left[2,m_{max}\right],\;n\in [1,n_{max}]$). Subsequently, q and $c_{a}$ were calculated using the constrained least squares for the $k^{th}$ pair ${\left(m,n\right)}_{k}$, the sample points (x,y), and the polynomial intervals v. This was repeated with randomly selected v N times, and the case with the smallest $c_{a}$ was selected. If $c_{a}$ was greater than $\epsilon_{a}$, the next pair ${\left(m,n\right)}_{k+1}$ was examined and q and $c_{a}$ were calculated again. The loop was repeated until $c_{a}$ was less than $\epsilon_{a}$. When $c_{a}$ became smaller than $\epsilon_{a}$, the piecewise polynomials with the order and the number of intervals were determined as the optimized approximate. If the $c_{a}$ of the pair ${\left(m,n\right)}_{max}$ was greater than $\epsilon_{a}$ for the pair, it was considered that finding the piecewise polynomial approximation function failed in the specified range. In this case, it would be possible to continue searching by using larger $m_{max}$ and/or $n_{max}$ values, or by relaxing $\epsilon_{a}$.

### 3.2. Computational Cost

In this section, the computational cost is described. Since the approximated function was piecewise with several polynomials, at runtime, it took two computation times: the time taken for a single polynomial computation and the time taken for selecting a specific polynomial. Therefore, the computational cost function is defined as follows:(5)$$c_{c}\left(m,n\right)=c_{p}+c_{b}.$$

The computing time for a single polynomial depends on the order of polynomial. This is defined as $c_{p}\left(m\right),$ as follows:(6)$$c_{p}\left(m\right)=n_{a}r_{a}+n_{r}r_{r}m+n_{f}r_{f}m,$$
where $n_{a}$, $n_{r}$, and $n_{f}$ are the number of assignments, arithmetic, and ‘for’ instructions, respectively. In addition, $r_{a}$, $r_{r}$, and $r_{f}$ are cycle counts of assignment, arithmetic, and ‘for’ instructions. m, the order of polynomial, indicates the repetition count.

Since the approximate function is implemented with several polynomials, the clock cycles are required for the binary search, which is intended to find the suitable intervals for a certain input x. This is defined as $c_{b}$ and determined as the average of the short path case and long path case in a binary search tree. Two examples are illustrated in Figure 3.

Since the average clock cycles to determine the corresponding polynomial depend on the probability that the input is in a certain interval, $c_{b}$ is a probability distribution function for the input value. For the sake of simplicity, the probability that an input is in a particular interval is the same for all intervals. Consequently, the costs associated with the number of intervals are as follows:(7)$$c_{b}=c_{bs}p_{s}+c_{bl}p_{l},$$
where $c_{bs}=(d_{m}-1)(n_{a}r_{a}+n_{d}r_{d}+n_{r}r_{r}+n_{w}r_{w}+n_{i}r_{i})$, $p_{s}=n_{s}/n$, $c_{bl}=d_{m}(n_{a}r_{a}+n_{d}r_{d}+n_{r}r_{r}+n_{w}r_{w}+n_{i}r_{i})$, and $p_{l}=n_{l}/n$. $c_{bs}$ and $c_{bl}$ represent the binary search tree costs for the short and the long path, respectively. Similarly, $n_{s}$ and $n_{l}$ are the numbers of the short path and the long path, respectively. Note that $n_{s}+n_{l}=n$. $d_{m}$ indicates the depth of the tree and is an integer value satisfying $2^{d_{m}-1}\leq n<2^{d_{m}}$. In addition, $n_{d}$, $n_{h}$, and $n_{i}$ are the numbers of divisions, ‘while’, and ‘if’ instructions, respectively. Finally, $r_{d}$, $r_{h}$, and $r_{i}$ are cycle counts of division, ‘while’, and ‘if’ instruction for the ARM Cortex-M7 core (see [18]), respectively. The calculated $c_{c}(m,n)$ is organized into a table by m and n (see Figure 4) and sorted in ascending order. In this paper, it is a priority to increase m when the costs are the same.

### 3.3. Application of the OPP Approximation Algorithm

The approximation function determined through the OPP algorithm is applied to the system and used. Implementing the OPP algorithm has several advantages. First, the OPP algorithm improves the calculation process by minimizing the resources required for real-time calculations. Second, this speeds up the execution time and improves the overall performance of the system. It makes resource management easier by reducing memory and computational load in the system. This is especially important for efficiently using resources in systems with limited hardware capabilities. Third, the algorithm can process pre-computations depending on the size and complexity of the data, allowing for efficient expansion. Thanks to this scalability, the OPP algorithm is suitable for a wide range of applications across various domains. Including the OPP approximation algorithm in the system design not only simplifies the computational process, but also improves the system’s ability to efficiently manage resources and handle complex datasets.

## 4. Experimental Results

In order to verify the performance of the proposed OPP algorithm, it was implemented using MATLAB with several nonlinear functions. In this experiment, we set the number of samples to 100, $m_{max}=10$, $n_{max}=20$, and $\epsilon_{a}=0.0001$. And the boundary values were determined as the values when the approximate errors were the smallest after randomly changing 100 times. Figure 5 is the result of approximation for the logarithmic function. The logarithmic function was chosen as the test function because there are many sensors whose output results are logarithmic, such as temperature sensors and distance sensors. Figure 6 shows the approximation results for the sine function that is frequently used in many applications. Figure 7 and Figure 8 show the approximate results for the triangular and square functions, respectively. The triangular function is adopted to examine an approximation of the continuous nonlinear function with undifferentiable points. In Figure 7, the approximation is shown with a smooth curve at a sharp point. And the square function is used for curve fitting of the function, which has discontinuous points. Figure 8 shows that overfitting occurs in the section where the value changes rapidly, resulting in oscillation. Figure 9 is the result of approximation of the sigmoid function. The sigmoid function, also known as the logistic function, is commonly used in various fields such as neural networks, machine learning, and statistics. Figure 10 is the result of approximation for the ReLU function, which is one of the most commonly used activation functions in deep learning models. Figure 11 is the result of the user-specified function for testing a nonlinear function that is more complex than the previous differential functions across all intervals. The function is represented as follows:(8)$$y=\frac{1}{1+e^{-x}}+\sin\left(x\right)-\ln\left(x\right).$$

Overall, the approximation for continuous and differentiable functions over all intervals is a satisfactory result. However, in functions with continuous, but undifferentiable, points and functions with discontinuous points, they have lower accuracy and higher $c_{c}(m,n)$ than continuous functions. This can be solved by increasing the sample point. Another method is to vibrate the boundary value, which will be studied later.

## 5. Conclusions

To approximate the complex function or discrete sample points, the optimal piecewise polynomial (OPP) approximation algorithm was proposed. The maximum values of polynomial order and the number of intervals to explore were preset, and the computational cost was calculated to determine the order and number of intervals for the approximation function. Then, the combination of the order and the number of intervals was sorted in ascending order based on the computational cost offline. Subsequently, the coefficient of the polynomial and the approximation error at the sample points were determined using constrained least squares. If the error was greater than the given error bound, the next combination was examined until the error was less than the bound. Ultimately, the OPP approximation algorithm determined the fastest approximation function at a runtime within a permissible error margin. The performance of the proposed algorithm was demonstrated through several nonlinear functions.

There are some further works needed to obtain a more accurate approximation function. In this paper, the optimal intervals were obtaioned by random sampling, i.e., the Monte Carlo method. To improve the performance, the gradient-based method will also be applied to determine the optimal boundaries, which will minimize the approximation error. Moreover, the performance will be verified and analyzed in experiments with actual embedded systems.

## Figures and Tables

**Figure 1 sensors-24-03991-f001:**
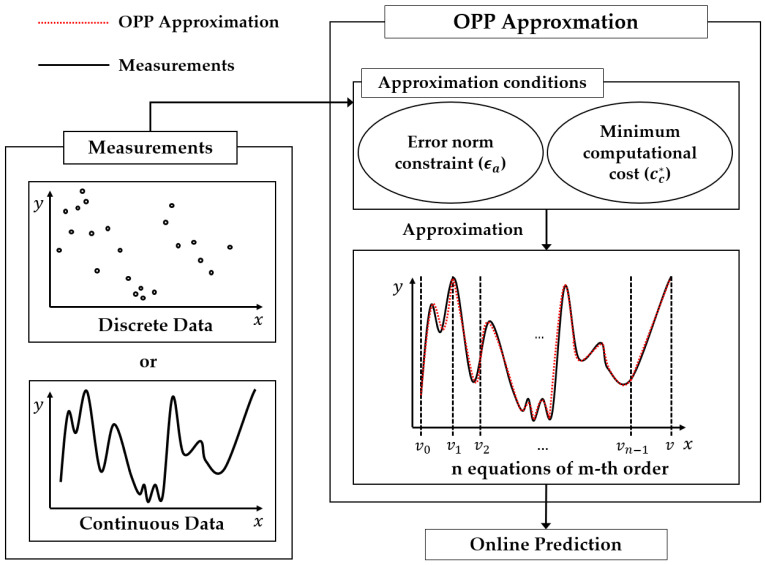
OPP approximation algorithm overview.

**Figure 2 sensors-24-03991-f002:**
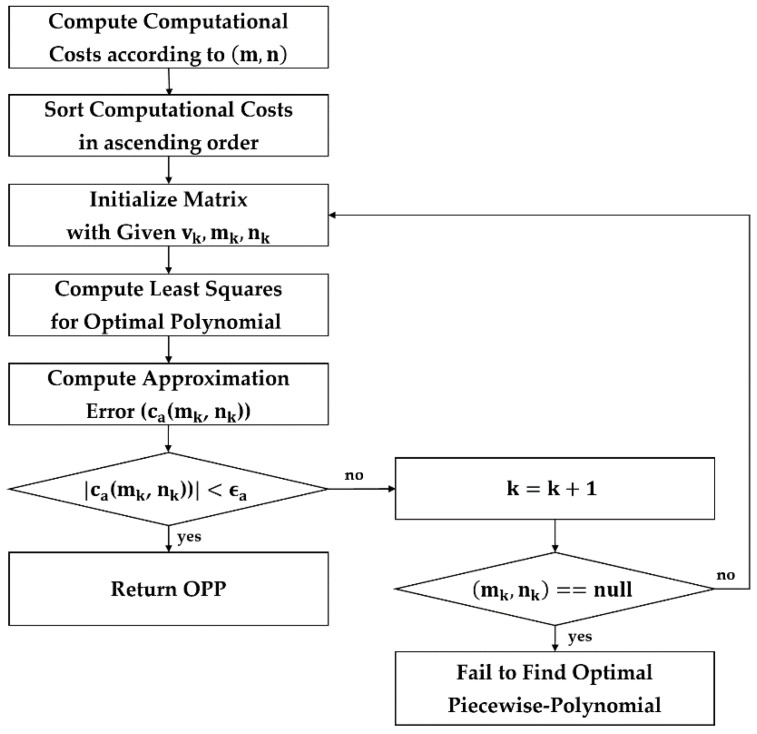
Overall algorithm flow for OPP approximation.

**Figure 3 sensors-24-03991-f003:**
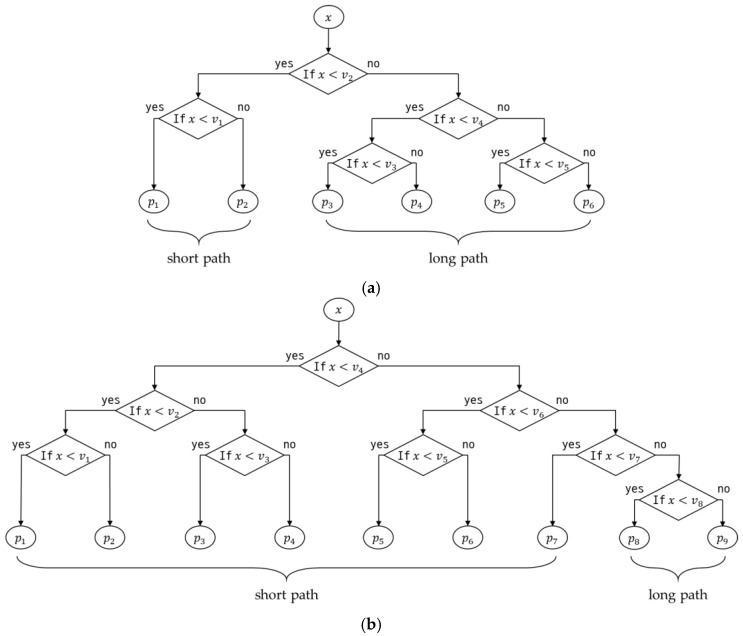
Binary search tree to select a suitable polynomial corresponding to input value. (**a**) Example 1: n=6. (**b**) Example 2: n=9.

**Figure 4 sensors-24-03991-f004:**
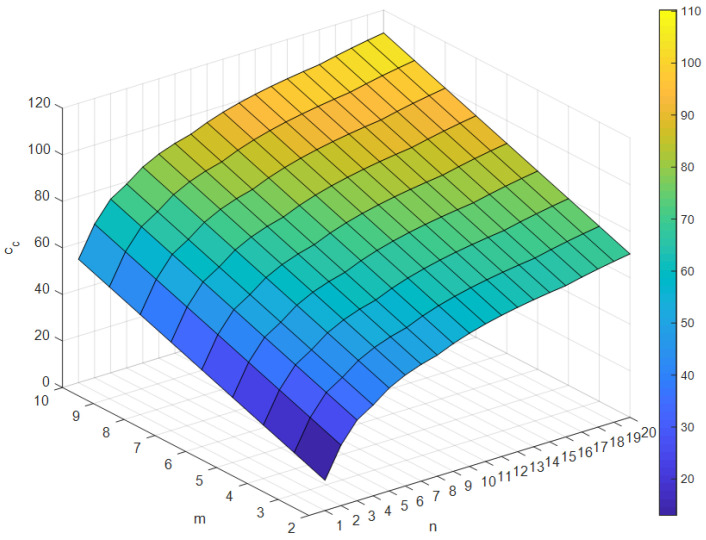
Computational cost according to the order of polynomial $\left(m\right)$ and the number of intervals (n)
$\left(m_{max}=10,\;n_{max}=20\right)$.

**Figure 5 sensors-24-03991-f005:**
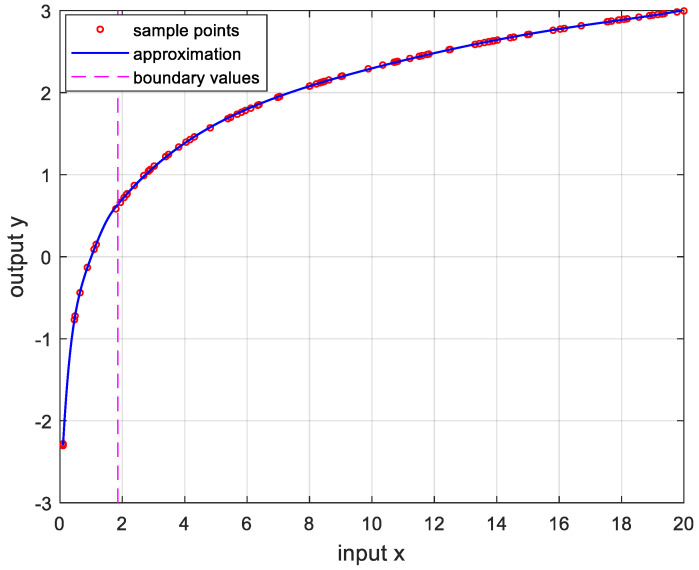
Approximation for $ln\left(x\right)$, $\left(\begin{array}{c} 0.1\leq x\leq \\ 20 \end{array}\right)$. $\left(m^{*},n^{*}\right)=\left(5,2\right),$ and the error is $3.3249\times 10^{-5}$.

**Figure 6 sensors-24-03991-f006:**
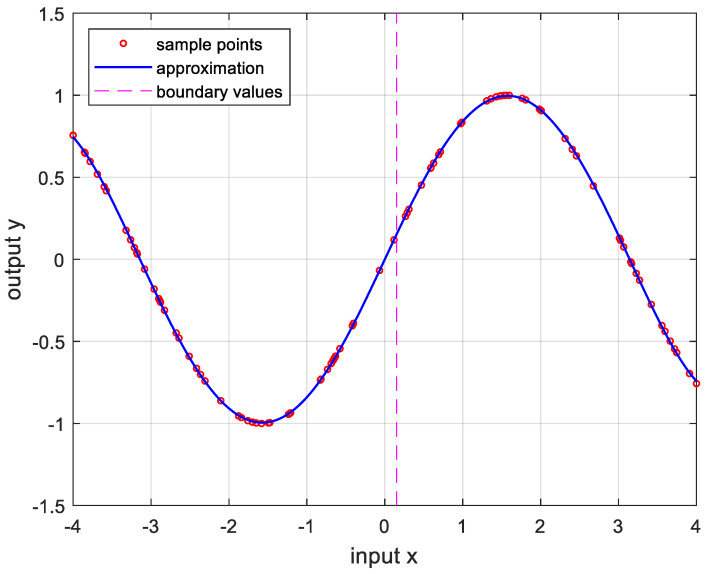
Approximation for $\sin\left(x\right)$, $\left(-4\leq x\leq 4\right)$. $\left(m^{*},n^{*}\right)=\left(4,2\right),$ and the error is $2.0771\times 10^{-5}$.

**Figure 7 sensors-24-03991-f007:**
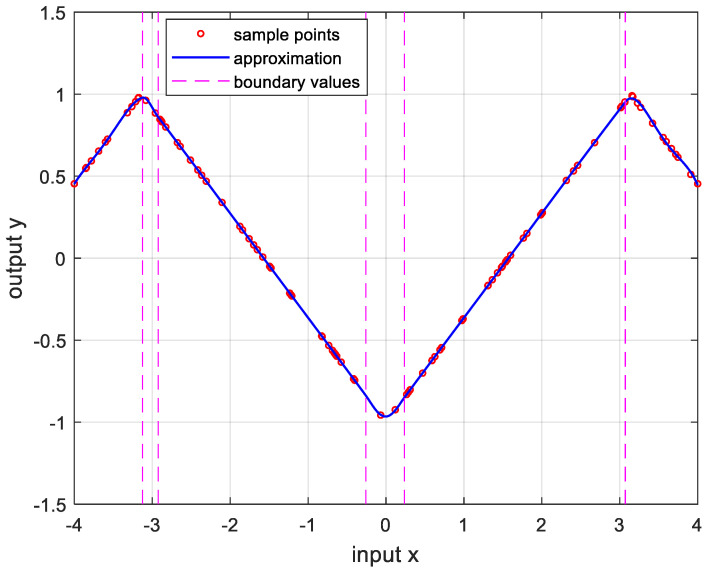
Approximation for $sawtooth\left(x\right)$, $\left(-4\leq x\leq 4\right)$. $\left(m^{*},n^{*}\right)=\left(4,6\right),$ and the error is $1.8994\times 10^{-5}$.

**Figure 8 sensors-24-03991-f008:**
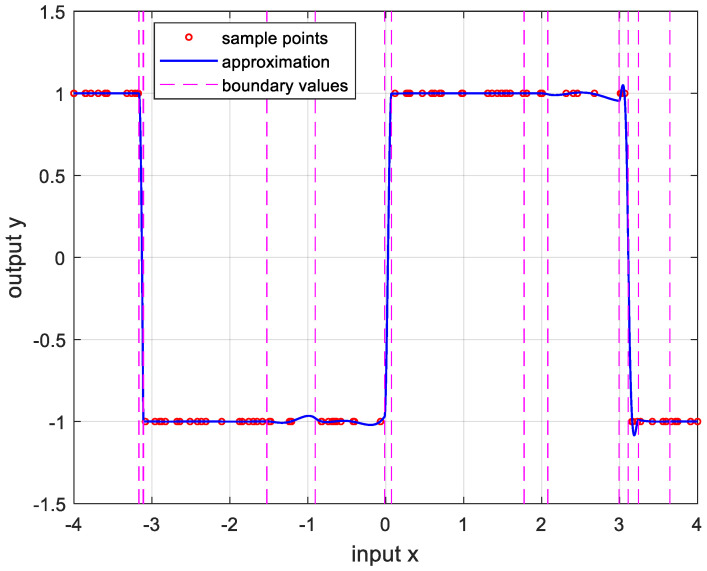
Approximation for $square\left(x\right)$, $\left(-4\leq x\leq 4\right)$. $\left(m^{*},n^{*}\right)=\left(4,14\right),$ and the error is ${5.4552\times 10}^{-5}$.

**Figure 9 sensors-24-03991-f009:**
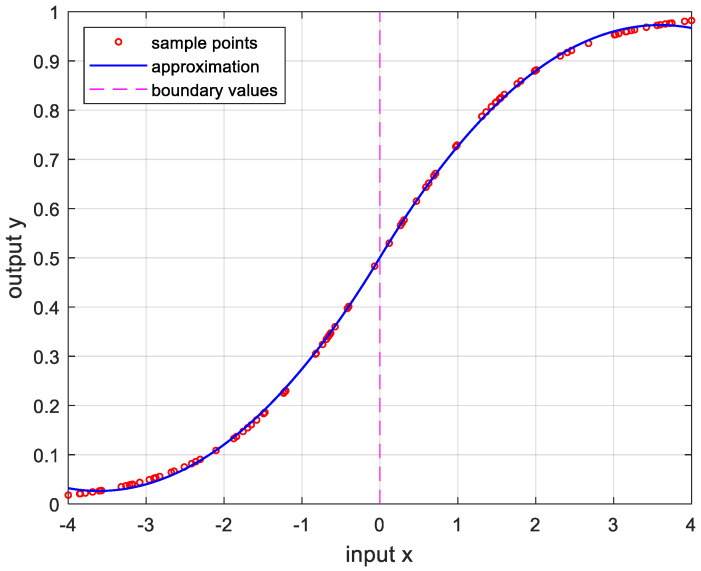
Approximation for $\frac{1}{1+e^{-x}}$, $\left(-4\leq x\leq 4\right)$. $\left(m^{*},n^{*}\right)=\left(2,2\right),$ and the error is ${2.8652\times 10}^{-5}$.

**Figure 10 sensors-24-03991-f010:**
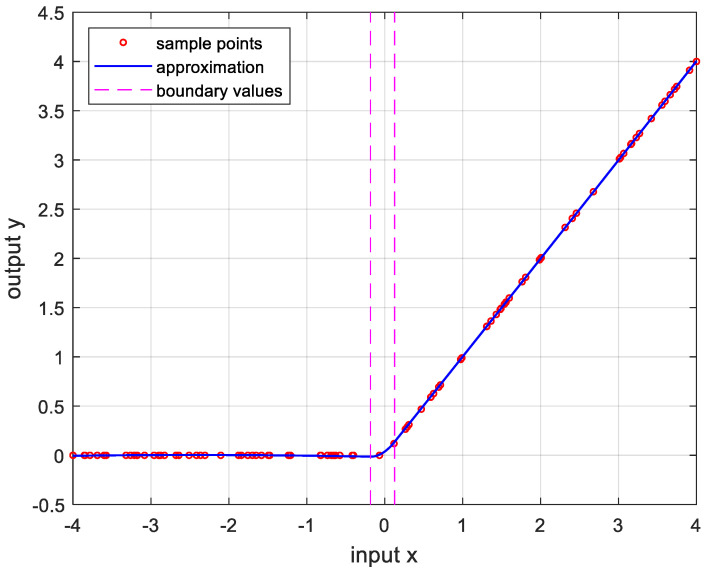
Approximation for $max\left(0,x\right)$, $\left(-4\leq x\leq 4\right)$. $\left(m^{*},n^{*}\right)=\left(2,3\right),$ and the error is ${1.5845\times 10}^{-5}$.

**Figure 11 sensors-24-03991-f011:**
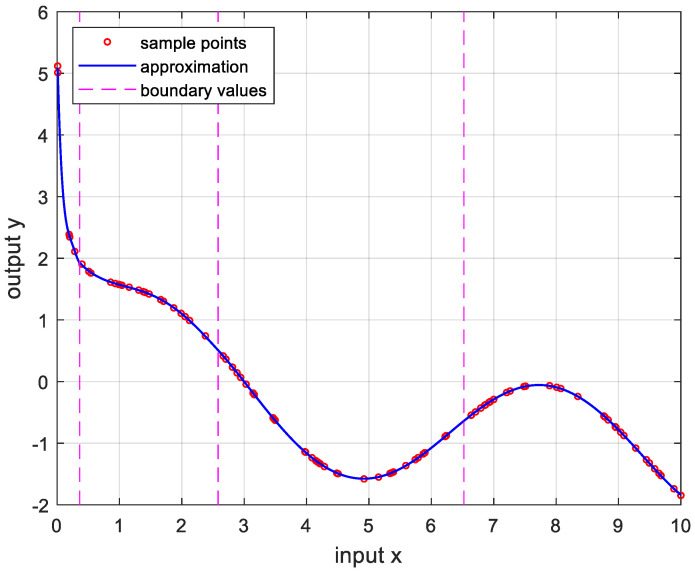
Approximation for (8), (0.1≤x≤10). $\left(m^{*},n^{*})=(4,4\right)$, and the error is ${2.9888\times 10}^{-5}$.

**Table 1 sensors-24-03991-t001:** Meanings of symbols.

Symbol	Signification
$f\left(x\right)$	Function to approximate
$x_{k,j}$	$\mathrm{The}\;j^{th}$$\;\mathrm{sample}\;\mathrm{in}\;\mathrm{the}\;k^{th}$ interval
$y_{k,j}$	$\mathrm{The}\;j^{th}$$\;\mathrm{function}\;\mathrm{value}\;\mathrm{in}\;\mathrm{the}\;k^{th}$ interval
η(k)	$\mathrm{Number}\;\mathrm{of}\;\mathrm{samples}\;\mathrm{for}\;\mathrm{the}\;k^{th}$ interval
m	Polynomial order
n	Number of intervals
$q_{k}$	(m+1)×1$\;\mathrm{vector}\;\mathrm{of}\;\mathrm{polynomial}\;\mathrm{coeffs}\;\mathrm{in}\;\mathrm{the}\;k^{th}$ interval
q	(m+1)n×1 $\;\mathrm{vector}\;\mathrm{concatenated}\;\mathrm{all}\;q_{k}$
$a_{k,i}$	$\mathrm{Coefficient}\;\mathrm{of}\;\mathrm{the}\;i^{th}$$\;\mathrm{order}\;\mathrm{term}\;\mathrm{for}\;\mathrm{the}\;k^{th}$ polynomial
v	(n+1)×1 vector of boundary values
$c_{a}$	Average sum of error squares at sample points
$\epsilon_{a}$	Error norm constraint for approximation
$c_{c}(m,n)$	Computational cost according to (m,n) polynomial expressed by CPU instruction cycles
$\left(m^{*},n^{*}\right)$	(m,n) $\;\mathrm{for}\;\mathrm{the}\;\mathrm{minimal}\;\mathrm{computational}\;\cos t\;\mathrm{with}\;|c_{c}(m,n)|\leq \epsilon_{a}$
$p_{k}$	$\mathrm{Approximation}\;\mathrm{polynomial}\;\mathrm{for}\;\mathrm{the}\;k^{th}$ interval

## Data Availability

No new data were created or analyzed in this study. Data sharing is not applicable to this article.

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
