# Peer review of "Optimal Piecewise Polynomial Approximation for Minimum Computing Cost by Using Constrained Least Squares"

_sensors, 2024, doi:10.3390/s24123991_

Round 1

Reviewer 1 Report

Comments and Suggestions for Authors

The paper presents an optimal approximation algorithm aimed at simplifying non-linear functions and discrete data into piecewise polynomials using constrained least squares. The focus on reducing computational costs while maintaining accuracy in time-sensitive and resource-limited environments is highly relevant. This addresses a critical need in embedded systems and real-time applications.
The paper clearly states the objective of finding the Optimal Piecewise Polynomial (OPP) with minimum computational cost while ensuring that the error remains below a specified threshold. This clarity helps in understanding the goals of the research.

While the paper discusses computational costs, it does not delve deeply into the scalability of the algorithm. For large-scale data sets or more complex functions, the feasibility of the approach may be challenged. More discussion on how the algorithm scales with increased data size or function complexity would be beneficial.

Reviewer 2 Report

Comments and Suggestions for Authors

The work contains a lot of information. Changes are necessary:

My comments:

Editorial errors – check the editorial style and MDPI requirements.

The summary does not reflect the entire article. Show the newness and differences of the ten articles and the studies in it from other studies.

In the introduction, add sections that contain the substance of what is being done in the article.

There is no research methodology – add a research methodology section and include it in other research questions.

How can a new algorithm be different, what is different? There is a lack of literature analysis on this topic.

The application section is too short. It needs to be developed.

Add new research studies to your references.

Reviewer 3 Report

Comments and Suggestions for Authors

This paper presents an optimal least-squares based interpolation methodology to approximate functions based on a set of random samples. The method considers the computational cost of the polynomial approximation as well as an iterative method to minimize the error with prospective applications for embedded systems:

1) The sum should be divided by n in equations (1) to (3) 

2) What is the effect on the function estimation when gaussian noise is added to the random samples taken from the function? How robust is the least squares aproach.

3) In this case, given that the functions are well defined the offline least squares performs properly. However, in the case of a sensor function approximation, a recursive approach for least-squares may be required to account for the random noise and variance on the estimation.

3) please state clear on the introduction the contributions of this manuscript as well as define why the offline least squares can be considered in this case.

4) Can you provide a microcontroller based execution to evaluate the effectiviness of this method. One alternative could be using code generation with an arduino or any board that supports matlab code generation.

5) please revise the manuscript for typos. e.a in the conclusions, "the i.e. Mote Carlo method" should be " Monte Carlo 260 method." 

Round 2

Reviewer 3 Report

Comments and Suggestions for Authors

The authors addressed reviewers comments, it can be accepted for publication